# High-Strength Building Material Based on a Glass Concrete Binder Obtained by Mechanical Activation

Sergey S. Dobrosmyslov [1,2,*], Vladimir E. Zadov [1], Rashit A. Nazirov [2], Veronika A. Shakirova [2], Anton S. Voronin [1,2], Michail M. Simunin [1,2], Yuri V. Fadeev [1,2], Maxim S. Molokeev [3,4], Ksenia A. Shabanova [1] and Stanislav V. Khartov [1]

[1] Federal Research Center Krasnoyarsk Scientific Center, Siberian Branch, Russian Academy of Sciences (FRC KSC SB RAS), 660036 Krasnoyarsk, Russia; a.voronin1988@mail.ru (A.S.V.); daf.hf@list.ru (Y.V.F.)

[2] School of Engineering and Construction, Siberian Federal University, 660041 Krasnoyarsk, Russia; nazirovra@gmail.com (R.A.N.)

[3] Laboratory of Crystal Physics, Kirensky Institute of Physics, Federal Research Center, Siberian Branch, Russian Academy of Sciences, 660036 Krasnoyarsk, Russia; msmolokeev@mail.ru

[4] Laboratory of Theory and Optimization of Chemical and Technological Processes, University of Tyumen, 625003 Tyumen, Russia

* Correspondence: dobrosmislov.s.s@gmail.com; Tel.: +7-906-9158-988

**Abstract:** As part of the work, the chemical interaction of finely ground glass (~1 μm), calcium oxide, and water was studied. It is shown that an increase in the fineness of grinding makes it possible to abandon autoclave hardening in the production of products on a hydrosilicate binder. The study of chemical interaction was carried out by calculating the thermodynamic equilibrium and was also confirmed by XRD analysis. DTA analysis showed that an increase in the treatment temperature leads to an increase in the proportion of the reacted phase at the first stage. Subsequently, phase formation is associated with the presence of CaO. The carrier of strength characteristics is the $CaO \times 2SiO_2 \times 2H_2O$ phase. The selection and optimization of the composition make it possible to obtain a high-strength glass concrete material with a strength of about 110 MPa. The micrographs of the obtained samples correspond to classical hydrosilicate systems.

**Keywords:** glass; building materials; mechanical activation; waste processing

## 1. Introduction

In the process of its development, mankind produces a large amount of non-decomposable and difficult-to-decompose materials [1], after which a large amount of waste is generated that has a significant anthropogenic load on the environment [2]. The disposal of these wastes in landfills is not rational from an economic point of view [3], as it removes the material from economic circulation. The use of waste in the production of materials will reduce the anthropogenic load on the environment and reduce the cost of finished products. The most typical example is industrial and domestic glass waste [4,5]. From an environmental point of view, glass is considered the most difficult waste to dispose of. It is not subjected to destruction under the influence of water, the atmosphere, solar radiation, or frost. In addition, glass is a corrosion-resistant material that does not collapse under the influence of an overwhelming amount of strong and weak organic, mineral, and bioacids, salts, as well as fungi and bacteria. Therefore, if organic waste completely decomposes after 1–3 years and polymeric materials after 5–20 years, then glass can be preserved without much damage for tens and even hundreds of years.

A large amount of industrial and domestic glass waste is generated annually in the world, which can potentially be used in the production of building materials and products [6]. In addition, the use of waste is 2–3 times cheaper than natural raw materials [7],

fuel consumption when using certain types of waste is reduced by 10–40%, and specific capital investments by 30–50%. At the same time, the use of glass as a filler for concrete compositions is difficult due to the softening of the system during the chemical interaction of glass and cement, respectively, the strength of such a product is not high [8]. In general, the use of glass as a filler for various concrete compositions is a promising direction. [9].

Glass contains silicon oxide (~72 wt%) [10] (Table 1), which, according to thermo-dynamics [11], should enter into a chemical reaction with calcium oxide to form calcium silicates, which act as a binder in concrete compositions [12]. The use of calcium oxide and silicon oxide as binder components makes it possible to reduce the amount of cement or exclude cement from the composition [13].

However, today the strength of cementless binders [14–16] remains low, about 20–30 MPa in compression. High-strength materials can only be obtained by autoclave hardening, which significantly increases the cost and energy consumption [17]. Reducing the particle size leads to an increase in chemical activity. Accordingly, a high fineness of grinding increases the chemical activity of the system, which makes it possible to obtain high-strength material without the use of an autoclave.

An increase in the specific surface area significantly increases the water demand during mixing, which in turn leads to a decrease in the physical and mechanical characteristics of glass concrete. This problem can be solved by using hyperplasticizers [18]. A decrease in the average particle size leads to an increase in the water demand of the material, which negatively affects the strength of the system. The use of hyperplasticizers can reduce the material's water demand. Therefore, an important aspect of this work is the correct combination of fine and coarse glass. Proper selection of these components allows you to obtain high-strength material.

Reducing the consumption of cement significantly reduces energy consumption in the production of building products as well as $CO_2$ emissions [19]. The combination of finely ground glass powder (obtained at a planetary mill), lime, and superplasticizers (reducing water demand) allows you to obtain a high-strength glass concrete composition.

As the main result of this work, it can be noted that the grinding of particles to a size of ~1 μm makes it possible to obtain a high-strength building material based on glass waste without the use of an autoclave.

**Table 1.** Equipment used.

| № | Denotation | Characterization |
|---|---|---|
| 1 | RETSCH PM 400 MA(Germany) planetary ball mill. | Specifications: 4 grinding jars with a nominal volume of 500 mL. The free speed setting from 30 to 400 rpm in combination with an effective 300 mm diameter planetary disk produces a particularly high transmitted energy. Speed ratio −1:−3. Ability to calculate the total energy transferred to the material during the grinding process. |
| 2 | FRITSCH ANALYSETTE 22 MicroTec PLUS(Germany) laser particle size analyzer | Measurement duration 5–10 s (registration of measurement results of one individual measurement), 2 min (full measurement cycle). Adjustable centrifugal pump with a maximum flow of 5.5 L/min. |
| 3 | Hitachi TM 4000 microscope (Japan) equipped with a BruckerXFlash 430 (Germany) X-ray microanalysis attachment | Zoom ×10–×100,000; Accelerating voltage: 5 kV, 10 kV, 15 kV; Table movement: X ± 40 mm, Y ± 35 mm; Maximum sample size: 80 mm in diameter, 50 mm in height; Minimum travel step: 65 nm; Detectors: 4-segment highly sensitive semiconductor detector, secondary electron detector for low vacuum mode; Electron source: pre-centered tungsten cathode; Country of origin: Japan. |
| 4 | Haoyuan DX-2700BH(China) powder diffractometer (analytical equipment of Krasnoyarsk Regional Center of Research Equipment of Federal Research Center "Krasnoyarsk Science Center SB RAS") | Cu-K$\alpha$ radiation and linear detector. The step size of 2θ was 0.01°, and the counting time was 0.2 s per step. Therefore, these structures were taken as starting model for Rietveld refinement which was performed using TOPAS 4.2 |
| 5 | Netzsch TG-DSK STA 409 PC Jupiter (Germany). | Sensitivity: 0.002 mg; operating temperature range: 25–1500 °C; rate of temperature change: 0.1–50 °C/min; the ability to work in neutral and oxidizing gas environments. |
| 6 | Instron 3360(USA) Series Testing Press (5 tons), for physico-mechanical tests. | Manufacturer information: Instron, USA. Purpose: Testing of building materials. Scope: Determination of strength characteristics of materials. Characteristics: Maximum load 5 tons. |

## 2. Materials and Methods

For the manufacture and study of glass concrete, the following equipment was used.

Compressive and flexural strengths were tested according to the method described in C109/C109M 13 Standard Test Method for Compressive Strength of Hydraulic Cement Mortars. An Instron 3369 testing machine was used.

Calcium oxide (chemically pure grade) was preliminarily fired at a temperature of 950 °C in order to dissociate $CaCO_3$ and $Ca(OH)_2$, possibly formed during storage. Waste window glass was used as cullet. Melflux 1641F was used as a superplasticizer, which was introduced by mixing water with 1% by weight of CaO.

To assess the chemical reactions in the binder, a mixture of 75% finely ground glass and 25% CaO was used, while the superplasticizer was not used to exclude its effect on chemical transformations. Previously, for the manufacture of composite concrete material, the following components were obtained:

Component №1-GP1 (glass power)

Broken sheet glass was ground in a ball mill to a powdery state with a particle size of the order of 50–60 microns (residue on a sieve > 66 microns is 60%).

Component №2-GP2 (glass power)

The resulting glass powder, with a particle size of 50–60 μm, was jointly ground with pre-fired calcium oxide to a final fineness of ~1 μm. In this study, sheet glass was used due to the stability of the chemical composition of Table 2.

**Table 2.** Chemical composition of glass.

| Glass | Mass Content, % | | | | | | |
|-------|------|------|------|------|------|------|------|
| | $SiO_2$ | $Al_2O_3$ | MgO | CaO | $Na_2O$ | $Fe_2O_3$ | $SO_3$ |
| sheet | 70.64 | 0.68 | 3.55 | 9.93 | 13.66 | 0.18 | 0.5 |

The synthesis of the material was carried out according to the scheme shown in Figure 1.

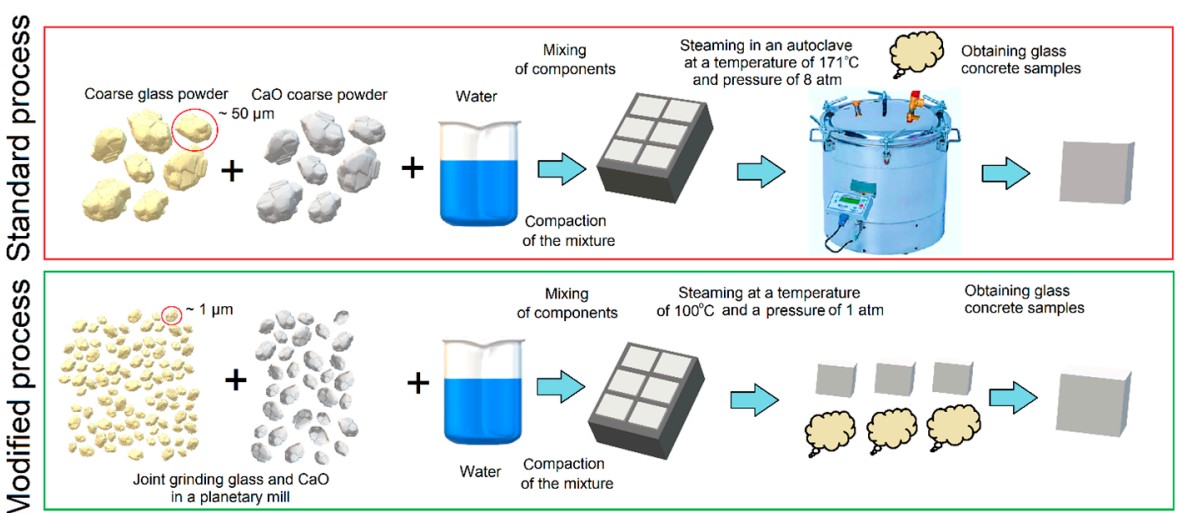

**Figure 1.** Schematic of the standard (autoclave) process for manufacturing products and the proposed scheme for manufacturing products with material grinding.

Samples were made in two ways. First, evaluate the chemical interaction in the binder based on Glass—CaO. In this case, the components (finely ground glass and chemically pure CaO) were mixed in a dry state, and water was added—20% of the total mass. Further, the mixture was mixed and vibrocompacted. Hardening took place in two stages: exposure at a temperature of 40 °C and steaming at 100 °C. Temperature largely determines the nature of

the chemical reaction of hydration [20]. At 20 °C, the mixture did not set, so the temperature was raised to 40 °C. The temperature of 100 °C simulates the process of steaming in industrial steam chambers. To assess the influence of the autoclave manufacturing method, the second stage was replaced by autoclave hardening at 180 °C and 8 atm. The choice of temperature and pressure is due to the presence of industrial autoclaves operating under these conditions.

The second method of manufacturing is associated with obtaining high-strength compositions. For this, the components GP1 and GP2 were mixed in specified proportions and hardened at 40 °C and 100% humidity for 24 h. Subsequently, the samples were steamed for 16 h at 100 °C.

## 3. Results

### 3.1. Study of the Chemical Interaction between Finely Ground Glass and Calcium Oxide

When studying chemical transformations, it was assumed that in an aqueous medium, only sodium silicates and silicon dioxide, which are part of the glass, enter into a chemical bond with calcium hydroxide; the remaining components do not enter into the reaction (they can only act as fillers). The reaction, when mixed with water, can proceed according to the following mechanism:

Fast reaction: $CaO + H_2O = Ca(OH)_2$.

The relatively slow reaction of the formation of calcium hydrosilicates:

$$aCa(OH)_2 + bNa_2SiO_3 + cSiO_2 + dH_2O = aCaO \times (b+c)SiO_2 \times (a+d-b)H_2O + 2bNaOH \quad (1)$$

There are a, b, c, d—coefficients associated with the formation of various types of calcium hydrosilicates.

There are various phases of calcium hydrosilicates. We will calculate the thermodynamic equilibrium; the thermodynamic characteristics are presented in Table 3. As part of the work, the following phases were considered $3CaO \times SiO_2$, $2CaO \times SiO_2 \times H_2O$, $Ca(OH)_2$, $2CaO \times SiO_2$, $CaSiO_3$, $SiO_2$, $2CaO \times SiO_2 \times 1.167H_2O$, $2CaO \times 3SiO_2 \times 2.5H_2O$, $3CaO \times 2SiO_2 \times 3H_2O$, $4CaO \times 3SiO_2 \times 1.5H_2O$, $5CaO \times 6SiO_2 \times 3H_2O$, $5CaO \times 6SiO_2 \times 5.5H_2O$, $5CaO \times 6SiO_2 \times 10.5H_2O$, $6CaO \times 6SiO_2 \times H_2O$ и $CaO \times 2SiO_2 \times 2H_2O$.

**Table 3.** Thermodynamic parameters of phases used in the calculation of thermodynamic equilibrium [21,22].

| № | Chemical Formula | S, J/mol·K | $\Delta H^o_{293}$, J/mol |
|---|---|---|---|
| 3 | $3CaO \times SiO_2$ | 168.6 | −2929.2 |
| 4 | $2CaO \times SiO_2 \times H_2O$ | 171.13 | −3138.84 |
| 5 | $Ca(OH)_2$ | 83.4 | −985.9 |
| 6 | $2CaO \times SiO_2$ | 120.8 | −2315.22 |
| 7 | $CaSiO_3$ | 171.13 | −3138.84 |
| 8 | $SiO_2$ | 41.46 | −910.86 |
| 9 | $2CaO \times SiO_2 \times 1.167H_2O$ | 160.67 | −2665.21 |
| 10 | $2CaO \times 3SiO_2 \times 2.5H_2O$ | 271.54 | −4920.38 |
| 11 | $3CaO \times 2SiO_2 \times 3H_2O$ | 312.13 | −4782.31 |
| 12 | $4CaO \times 3SiO_2 \times 1.5H_2O$ | 330.33 | −6020.78 |
| 13 | $5CaO \times 6SiO_2 \times 3H_2O$ | 513.17 | −9935.74 |
| 14 | $5CaO \times 6SiO_2 \times 5.5H_2O$ | 611.49 | −10,686.77 |
| 15 | $5CaO \times 6SiO_2 \times 10.5H_2O$ | 808.14 | −12,179.6 |
| 16 | $6CaO \times 6SiO_2 \times H_2O$ | 507.52 | −10,024.86 |
| 17 | $CaO \times 2SiO_2 \times 2H_2O$ | 171.13 | −3138.84 |

Let us assume that $H_2O$ enters the cluster consisting of $SiO_2$–$CaO$. Thus, we simulate the hydration process and calculate the thermodynamic equilibrium using the data presented in Table 3. We will carry out the calculation for the following temperatures: 40, 100, and 180 °C. 40 °C corresponds to the hydration process occurring at the first

stage of synthesis. 100 °C corresponds to the steaming process at atmospheric pressure. 180 °C—hydration in an autoclave was simulated at a pressure of 8 atm, which corresponds to an industrial autoclave. The results of the thermodynamic equilibrium calculation are shown in Figure 2.

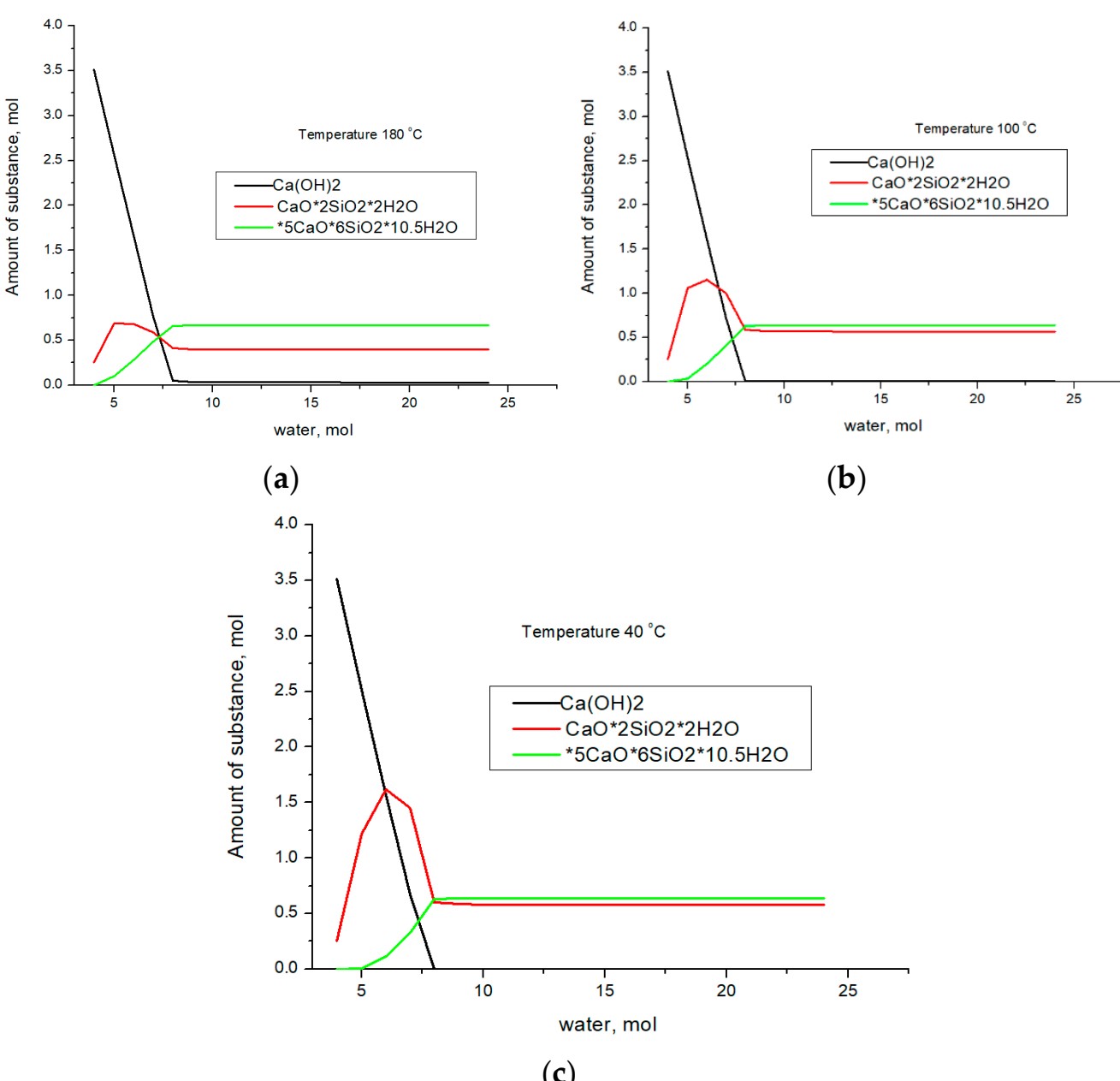

**Figure 2.** The results of calculating the thermodynamic equilibrium in the $SiO_2$–CaO–$H_2O$ system at 180 °C (**a**), 100 °C (**b**), and 40 °C (**c**).

As follows from the presented results, the formation of calcium hydroxide at the first stage of water entering the system is typical for all temperatures. Subsequently, the formation of the $CaO \times 2SiO_2 \times 2H_2O$ phase occurs; with an excess of water, the $5CaO \times 6SiO_2 \times 10.5H_2O$ phase begins to form. At the same time, the formation of $CaO \times 2SiO_2 \times 2H_2O$ should occur, despite the fact that, from a thermodynamic point of view, the formation of $Ca(OH)_2$ is more advantageous. A further increase in the amount of water in the system leads to the formation of $5CaO \times 6SiO_2 \times 10.5H_2O$, for which the ratio of calcium atoms to water molecules practically does not change; however, the ratio of silicon and water molecules increases from

1:1 to 1:1.75. Consequently, an increase in the water concentration in the system leads to the formation of the $5CaO \times 6SiO_2 \times 10.5H_2O$ compound from the $CaO \times 2SiO_2 \times 2H_2O$ phase, in which a larger amount of $SiO_2$ enters the reaction. The similarity of the presented curves indicates that phase transformations in the system are determined by the amount of water, while temperature and pressure are factors that accelerate the chemical reaction but do not determine the final phase composition. These phases act as a binder in the formation of the concrete composition. Let us compare the calculation data with the results of X-ray phase analysis. Figure 3 shows X-ray patterns for samples obtained at temperatures of 40, 100, and 180 °C for the composition of 75% finely ground glass and 25% CaO, and the figure also shows the results of quantitative analysis.

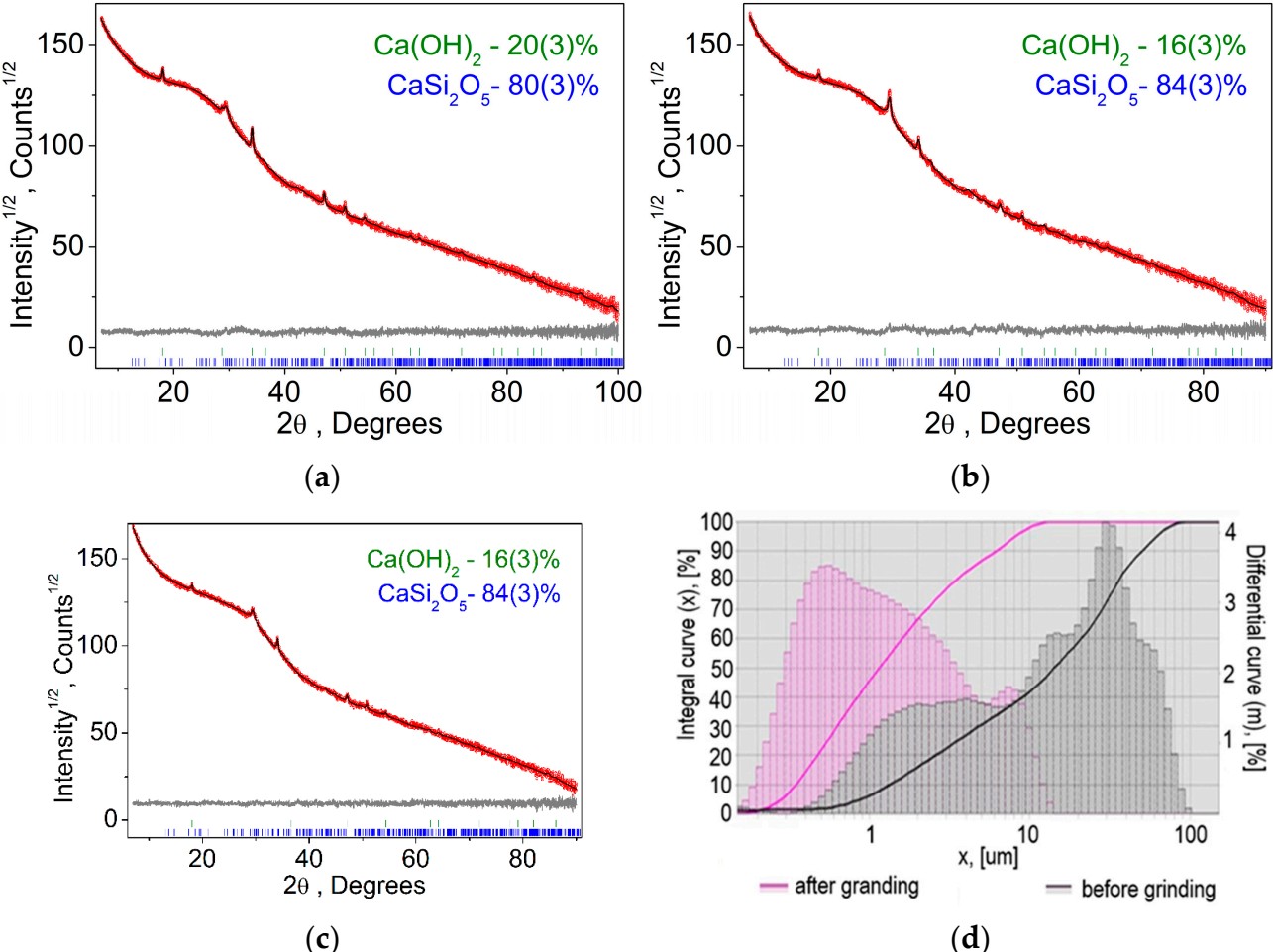

**Figure 3.** Difference Rietveld plot of glass concrete: (**a**) 40 °C; (**b**) 100 °C; (**c**) 180 °C; (**d**) results of granulometric analysis. The integral curve is the total amount of powder up to a given size, %. The differential curve is the amount of powder in a given size interval, %.

According to the results of X-ray phase analysis, the following results were obtained (for samples obtained at temperatures of 40, 100, and 180 °C). All peaks were indexed by two phases: $Ca(OH)_2$ and $CaSi_2O_5$ [23]. Refinements were stable and gave low R-factors (Table 4, Figure 3). This result confirms the thermodynamic calculations performed. The fact that only the $CaO \times 2SiO_2 \times 2H_2O$ phase was detected indicates that the formation of the $5CaO \times 6SiO_2 \times 10.5H_2O$ phase has not yet begun.

**Table 4.** Main parameters of processing and refinement of the samples of glass concrete at temperatures of 40, 100, and 180 °C.

| Synthesis Temperature | 40 °C | | 100 °C | | 180 °C | |
|---|---|---|---|---|---|---|
| **Phase** | **Ca(OH)$_2$** | **CaSi$_2$O$_5$** | **Ca(OH)$_2$** | **CaSi$_2$O$_5$** | **Ca(OH)$_2$** | **CaSi$_2$O$_5$** |
| Weight, % | 20(3) | 80(3) | 16(3) | 84(3) | 16(3) | 84(3) |
| Sp. Gr. | *Pm*-31 | *P*-1 | *Pm*-31 | *P*-1 | *Pm*-31 | *P*-1 |
| a (Å) | 3.5907(16) | 7.17(1) | 3.587(4) | 7.26(1) | 3.593(2) | 7.49(1) |
| b (Å) | 3.5907(16) | 7.67(1) | 3.587(4) | 7.64(1) | 3.593(2) | 7.45(1) |
| c (Å) | 4.917(3) | 6.573(9) | 4.884(6) | 6.538(9) | 4.913(4) | 6.52(1) |
| α (°) | 90 | 82.1(1) | 90 | 81.95(7) | 90 | 82.1(1) |
| β (°) | 90 | 85.4(1) | 90 | 85.25(7) | 90 | 84.6(1) |
| γ (°) | 120 | 68.7(1) | 120 | 69.04(6) | 120 | 70.1(1) |
| V (Å$^3$) | 54.90 (6) | 333.5(9) | 54.4(1) | 335.1(9) | 54.94(9) | 338.4(9) |
| R$_B$, % | 0.77 | 0.81 | 0.98 | 0.99 | 0.45 | 0.36 |
| 2θ-interval, ° | 10–100 | | 10–90 | | 10–90 | |
| R$_{wp}$, % | 2.39 | | 2.45 | | 2.10 | |
| R$_p$, % | 1.49 | | 1.56 | | 1.25 | |

The appearance of the synthesized samples is shown in Figure 4.

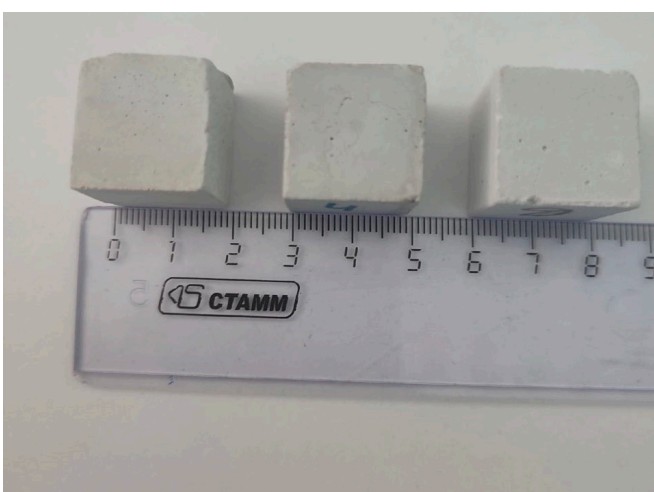

**Figure 4.** Appearance of samples.

As follows from the given X-ray patterns, two Ca(OH)$_2$ phases and the main binder phase CaO×2SiO$_2$ are formed during the synthesis, which are in good agreement with the results of thermodynamic calculations presented in Figure 2. This binder phase is described in the literature [24,25] and is formed in the presence of Na$_2$O, which is contained in glass. The hydration process proceeds according to the following mechanism. At the beginning, Na$_2$SiO$_3$ dissolves in a saturated Ca(OH)$_2$ solution with an excess of SiO$_2$, calcium hydrosilicates CaO×2SiO$_2$×2H$_2$O and 5CaO×6SiO$_2$×10.5H$_2$O are formed, and NaOH is formed, which interacts with glass to form Na$_2$SiO$_3$. Therefore, NaOH acts as a catalyst for the reaction. In this case, the reaction rate of the formation of Na$_2$SiO$_3$ is determined by the size of the diffusion barrier between the alkaline solution and the glass surface containing an excess amount of SiO$_2$. The chemical reaction scheme is given above in equation (1) and below in Equations (2) and (3). Free sodium hydroxide reacts with excess SiO$_2$, which is part of the glass, to form soluble sodium silicate:

$$SiO_2 + 2NaOH = Na_2SiO_3 + H_2O \tag{2}$$

A solution of sodium silicate reacts rapidly with a solution of calcium hydroxide to form sparingly soluble calcium silicate and strongly alkaline NaOH:

$$Na_2SiO_3 + Ca(OH)_2 = CaSiO_3 \downarrow + 2NaOH \qquad (3)$$

NaOH is regenerated as a result of the above reactions and is a catalyst for the formation of sparingly soluble calcium silicate. Calcium silicates, which have minimal solubility in the described system and have the composition $CaO \times 2SiO_2 \times 2H_2O$, $5CaO \times 6SiO_2 \times 10.5H_2O$, etc., are formed as a result of hydrothermal processes with the participation of excess $SiO_2$. Their formation is due to the thermodynamic and kinetic properties of the system.

Due to the fact that the chemical interaction in a heterogeneous reaction is associated with particle sizes, granulometric studies were carried out. The results of granulometric studies are shown in Figure 3d.

As follows from the above data, initially the powder was a bimodal system consisting of CaO and $SiO_2$ particles. After grinding particles less than 1 micron in the composition of the order of 50%. Figure 5 shows the results of DSC and TG analysis for a material with a composition of 75% finely ground glass and 25% CaO obtained at temperatures of 40, 100, and 180 °C.

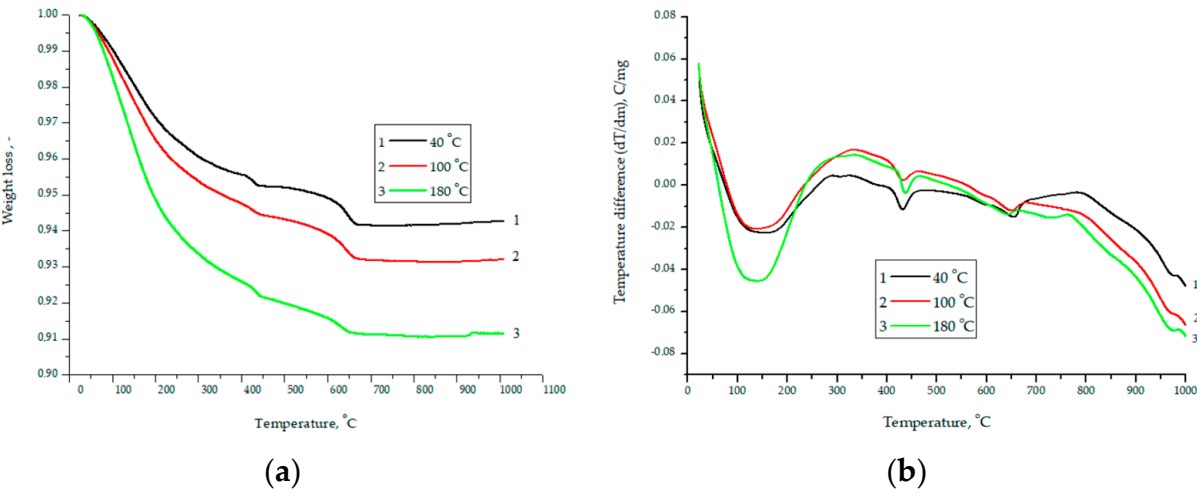

**Figure 5.** Data of TG (**a**) and DSC (**b**) analysis for the material composition of 75% finely ground glass and 25% CaO obtained at temperatures of 40, 100, and 180 °C.

With an increase in the hydration temperature, the total mass loss increases (Figure 5a). This result is due to the fact that with an increase in temperature, the rate of chemical reactions and the amount of hydrated components increase, and as a result, the mass loss increases. The smooth shape of curve-3 (Figure 5a) is explained by the fact that the degree of crystallinity increases during autoclaving, and the inflection at 650–670 °C is associated with the decomposition of calcium hydroxide [26]. As a characteristic difference between the DSC curves, it can be noted that a more pronounced endothermic peak at 150 °C is associated with the removal of physically adsorbed water. Therefore, at a pressure of 8 atm and a temperature of 180 °C, water enters the closed pore space. The difference in curve 1 after 650 °C is explained by the fact that capillary forces play a significant role in the process of structure formation. A further increase in the partial pressure of water vapor (at 1 atm 100 °C and 8 atm 180 °C) leads to the fact that the role of capillary forces becomes insignificant. essential. Figure 6 shows micrographs of the structure for the composition of 75% finely ground glass and 25% CaO obtained at temperatures of 40, 100, and 180 °C.

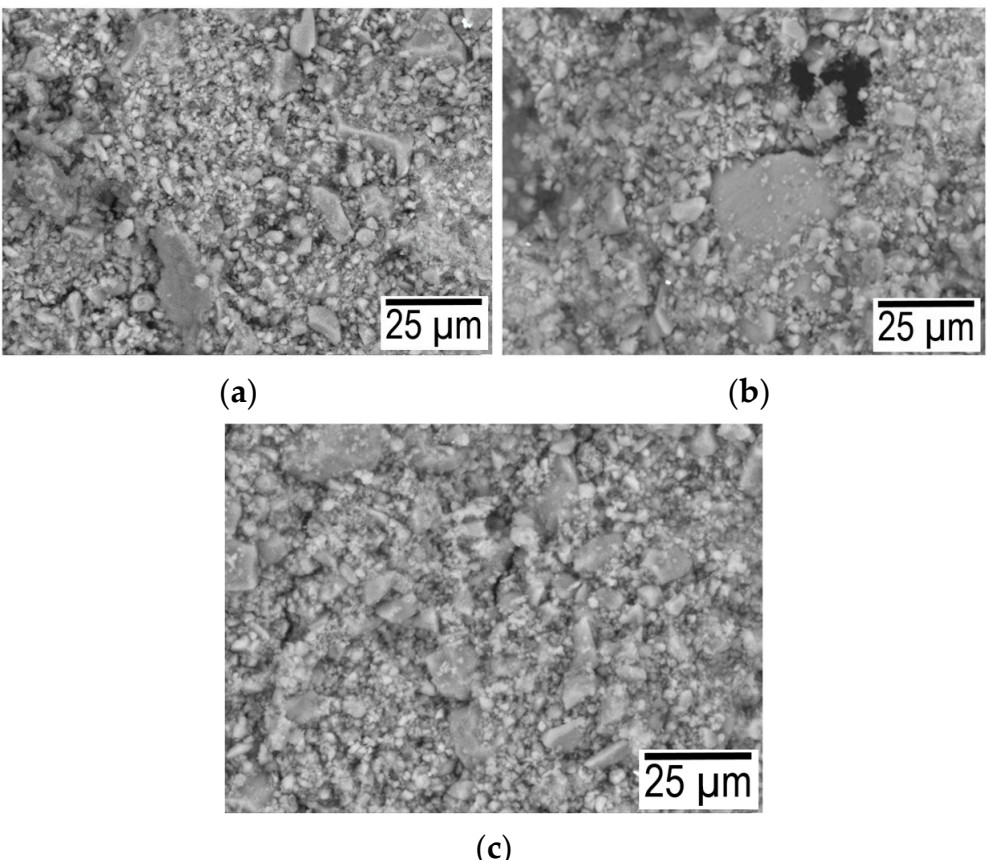

**Figure 6.** Micrographs for the composition of 75% finely ground glass and 25% CaO obtained at temperatures of 40 (**a**), 100 (**b**), and 180 (**c**) °C.

*3.2. Composite Material Based on a Glass Concrete Binder*

The method for preparing a glass mixture includes mixing the GP#1 component, the GP#2 component, the Melflux 1641 F plasticizer, and water using a concrete mixer until a homogeneous consistency is achieved. The resulting suspension was poured into molds and compacted on a vibrating table. To test the compressive strength, specimens with dimensions of 20 mm × 20 mm × 20 mm.

Hardening took place under various conditions:

(1)   In natural conditions (at a temperature of 40 ± 3 °C and a relative air humidity of 95 ± 5%) within 24 h;

(2)   Steaming samples at a temperature of 100 °C and a humidity of 100%.

Steaming was carried out with water vapor. According to the heat treatment time, the following modes were selected:

−   Short—1 and 4 h;

−   Normal—8 h;

−   Extended—16 h.

As part of the work, to determine the amount of the reacted phase, studies of the mass loss during TG analysis were carried out; the heating temperature was 1000 °C. Figure 7 shows the dependences of weight loss at various temperatures during thermogravimetric analysis for the compositions GP1/GP2: 60/40% and 80/20%.

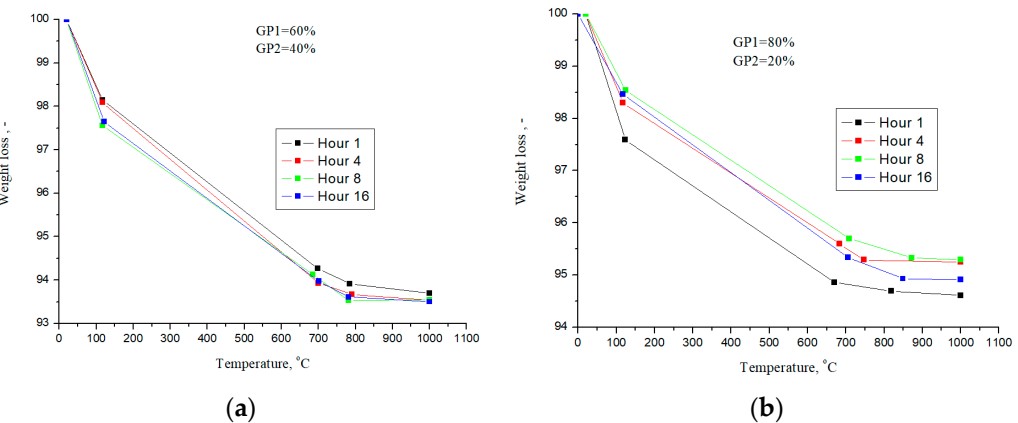

**Figure 7.** Weight loss during heat treatment for GP1/GP2 compositions—60/40% (**a**) and 80/20% (**b**).

Weight loss, for the GP1/GP2 sample—60/40%—during heat treatment is practically independent of the steaming time (Figure 7a). This effect is due to the fact that the $CaO \times 2SiO_2 \times 2H_2O$ and $5CaO \times 6SiO_2 \times 10.5H_2O$ phases are effectively formed, for which the amount of bound water is approximately the same. Insignificant differences in weight loss for the GP1/GP2 sample—80/20%, are due to the fact that a large amount of $Ca(OH)_2$ is formed at the first stage. Accordingly, 20% of GP2 powder is not enough to fully bind the lime to form calcium silicates.

A study of the morphology of the surfaces and chips of the samples showed that two types of particles were observed: agglomerates of smaller particles and lamellar formations. The results of the study are presented in Figure 8.

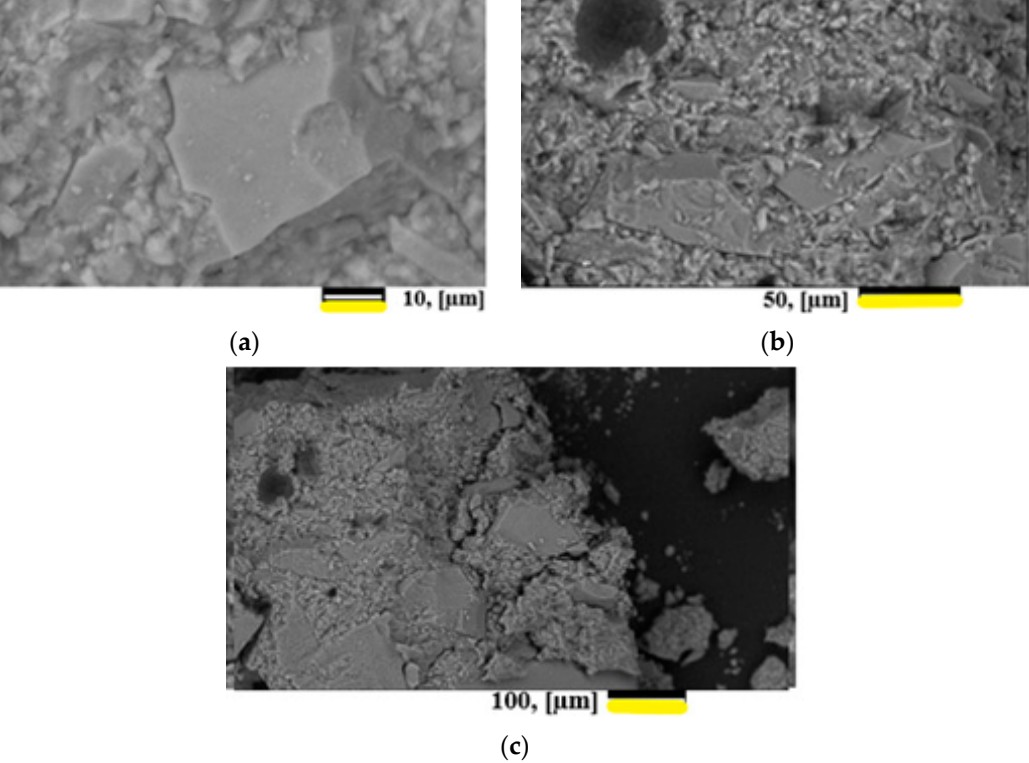

**Figure 8.** Micrographs of the cleavage surface of hardened materials for samples GP1/GP2—60/40%. (**a**) magnification 1.500 times. (**b**) magnification 500 times. (**c**) magnification 2500 times.

From the standpoint of the macrostructure, the hardened suspension is a composite material in which the role of the matrix is played by a colloidal glass suspension, and the filler is glass grains with gradient hydration of the layers.

Compressive Strength

For the compression test, cubes with dimensions of 20 mm × 20 mm × 20 mm were made, and the test was carried out for six samples per composition. Figure 9 shows compressive strength as a function of glass powder content with a final fineness of <1 μm (%) GP2 and steaming time.

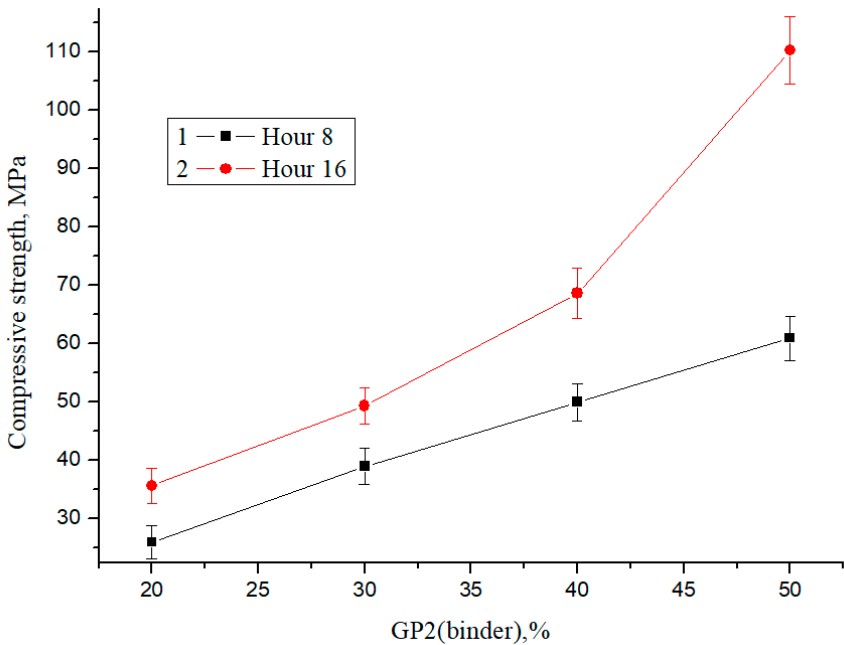

**Figure 9.** Dependence of the change in compressive strength on the content of binder powder GP2 and the heat treatment time.

As follows from the presented results, an increase in the amount of a finely dispersed phase leads to an increase in the relative increase in strength by 37% at a concentration of 20% binder and 81% at a concentration of 50% binder during steaming for 8 and 16 h, respectively. With an increase in the percentage of glass powder and a final fineness of grinding of <1 μm, the strength increases, and the dependence is linear. The maximum compressive strength is about 110 MPa; therefore, glass concrete can be considered a promising high-strength building material.

## 4. Conclusions

As part of the work, a high-strength building material was developed based on industrial and household glass waste and calcium oxide. As a binder in this material, finely ground glass (average particle size ~ 1 μm) and calcium oxide. Glass with dimensions of about 50–60 μm was used as a filler. As part of the work, it was shown that with an increase in the concentration of the binder (finely ground glass and calcium oxide) from 20 to 50%, the strength of the glass concrete composition increases from 35.7 to 110.3 MPa. Thermodynamic calculations and XRD analysis showed that hydrosilicate $CaO \times 2SiO_2 \times 2H_2O$ is formed during hydration, which is the main carrier of the strength properties of mixtures. An important factor influencing the process of phase formation of $CaO \times 2SiO_2 \times 2H_2O$ is the presence of NaOH in the glass. The results of DTA analysis and data from thermodynamic calculations show that an increase in temperature and pressure accelerates the process of phase formation of $CaO \times 2SiO_2 \times 2H_2O$, but the formation of other phases of calcium hydrosilicates does not occur.

**Author Contributions:** Conceptualization, S.S.D. and V.E.Z.; methodology R.A.N.; software, V.A.S.; validation, A.S.V. and Y.V.F.; formal analysis, V.E.Z.; investigation, M.S.M., K.A.S., and Y.V.F.; data curation, V.A.S.; writing—original draft preparation, S.S.D.; writing—review and editing, M.M.S.; visualization, A.S.V.; supervision, R.A.N.; project administration, S.V.K. All authors have read and agreed to the published version of the manuscript.

**Funding:** This research received no external funding.

**Informed Consent Statement:** Not applicable.

**Data Availability Statement:** Not applicable.

**Acknowledgments:** Studies electron microscopy were performed on the equipment of Krasnoyarsk. The Regional Center of Research Equipment of Federal Research Center «Krasnoyarsk Science Center SB RAS». This research did not receive any specific grant from funding agencies in the public, commercial, or not-for-profit sectors.

**Conflicts of Interest:** The authors declare no conflict of interest.

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
