# Peer review of "High-Strength Building Material Based on a Glass Concrete Binder Obtained by Mechanical Activation"

_buildings, doi:10.3390/buildings13081992_

Round 1

Reviewer 1 Report

Dear Authors, 

Thank you for submitting your manuscript. 

While the research topic is of greater interest, I believe, the methods and overall research design are not well explained in the manuscript. On several occasions the articles appears to be similar to a presentation slide with limited or no information at all. Majority of the results are presented without any explanation and seem to have no correlation with the earlier results (as presented in the manuscript). 

Please revise your manuscript with a proper and elaborate research design. 

Thank you. 

Moderate editing of English language required. 

Author Response

Thanks for the comment and more precise wording. 1 and 2 corrected. 3 and 4 - rewrote the paragraph.

Not high – low

Please revise this sentence.

Before:

The combination of finely ground glass powder (obtained at a planetary mill), lime and superplasticizers (reducing water demand) and allows you to get a high-strength glass concrete composition.

After:

A decrease in the average particle size leads to an increase in the water demand of the material, which negatively affects the strength of the system. The use of hyperplasticizers can reduce the water demand of the material. Therefore, an important aspect of this work is the correct combination of fine glass and coarse glass. Proper selection of these components allows you to get high-strength material.

Please revise this section with proper tables or paragraph. For the most part this section reads like a presentation slide.

Made it into a table.

The equipment used and its characteristics are given in the table

Table - characteristics of the equipment used

Denotation

Сharacterization

1

RETSCHPM 400 MA planetary ball mill, made in Germany

Specifications: 4 grinding jars with a nominal volume of 500 ml. The free speed setting from 30 to 400 rpm in combination with an effective 300 mm diameter planetary disk produces a particularly high transmitted energy. Speed ratio -1:-3. Ability to calculate the total energy transferred to the material during the grinding process.

2

FRITSCH ANALYSETTE 22 MicroTec PLUS laser particle size analyzer

Measurement duration 5–10 s (registration of measurement results of one individual measurement), 2 min (full measurement cycle). Adjustable centrifugal pump with a maximum flow of 5.5 l/min.

3

Hitachi TM 4000 microscope (Japan) equipped with a BruckerXFlash 430 (Germany) X-ray microanalysis attachment

Zoom x10 - x100 000; Accelerating voltage: 5 kV, 10 kV, 15 kV; Table movement: X±40 mm, Y±35 mm; Maximum sample size: 80 mm in diameter, 50 mm in height; Minimum travel step: 65 nm; Detectors: 4-segment highly sensitive semiconductor detector, secondary electron detector for low vacuum mode; Electron source: pre-centered tungsten cathode; Country of origin: Japan.

4

Haoyuan DX-2700BH powder diffractometer (analytical equipment of Krasnoyarsk Regional Center of Research Equipment of Federal Research Center "Krasnoyarsk Science Center SB RAS")

Cu-Kα radiation and linear detector. The step size of 2θ was 0.01°, and the counting time was 0.2 s per step. Therefore these structures were taken as starting model for Rietveld refinement which was performed using TOPAS 4.2

5

TG-DSK STA 409 PC Jupiter (Netzsch)

Sensitivity: 0.002 mg; operating temperature range: 25—1500 °С; rate of temperature change: 0.1–50 °C/min; the ability to work in neutral and oxidizing gas environments.

6

Instron 3360 Series Testing Press (5 tons)

Manufacturer information: Instron, USA. Purpose: Testing of building materials. Scope: Determination of strength characteristics of materials. Characteristics: Maximum load 5 tons.

Made in Germany or in Germany? Please clarify.

Fixed

Please tabulate this information or revise these sentences.

Added to table

Please revise this sentence

Added to table

Please put this information either in a table or revise the sentence.

Added to table

? – and

Please explain a little more about the results and trends.

Added text:

At the same time, the formation of CaO*2SiO2*2Н2O should occur, despite the fact that from a thermodynamic point of view, the formation of Ca(OH)2 is more advantageous. A further increase in the amount of water in the system leads to the formation of 5CaO*6SiO2*10.5H2O, for which the ratio of calcium atom to water molecules practically does not change, however, the ratio of silicon and water molecules increases from 1 : 1 to 1 : 1.75. Consequently, an increase in the water concentration in the system leads to the formation of the 5CaO*6SiO2*10.5Н2O compound from the CaO*2SiO2*2Н2O phase, in which a larger amount of SiO2 enters into the reaction. The similarity of the presented curves indicates that phase transformations in the system are determined by the amount of water, while temperature and pressure are factors that accelerate the chemical reaction and do not determine the final phase composition.

Please explain a little more about the results and trends.

Added text:

This result confirms the performed thermodynamic calculations. The fact that only the CaO*2SiO2*2Н2O phase was detected indicates that the formation of the 5CaO*6SiO2*10.5Н2O phase has not yet begun.

Please elaborate on the results. What is integral and what is differential?

Added a caption for the picture:

The integral curve is the total amount of powder up to a given size, %.

The differential curve is the amount of powder in a given size interval, %.

Rewrote the conclusion

Before:

The work carried out showed that the use of finely ground glass and calcium oxide obtained by mechanical activation in a planetary mill makes it possible to obtain a high-strength composite material. Based on the results of the work carried out, the following conclusions can be drawn:

  1. With an increase in the concentration of the binder (finely ground glass and calcium oxide) from 20 to 50%, the strength of the glass concrete composition increases from 35.7 to 110.3 MPa.
  2. According to the XPA results, the main component in the system is (C-S-H). The resulting hydrosilicate - CaO*2SiO2*2H2O is the main carrier of the strength properties of mixtures. The formation of this phase was confirmed by the study of thermodynamic equilibrium.
  3. The process of formation of the CaO*2SiO2*2H2O phase occurs according to the catalytic mechanism, where NaOH acts as a catalyst.
  4. According to the results of DTA analysis, an increase in the synthesis temperature leads to more intense phase formation, but in general, the grinding of particles to a size of ~ 1 μm is sufficient to activate chemical reactions.
  5. A study of the morphology of the surfaces and chips of the samples showed that 2 types of particles are observed: agglomerates of smaller particles and lamellar formations, which is typical for C-S-H systems.

After:

As part of the work, a high-strength building material was developed based on industrial and household glass waste and calcium oxide. As a binder in this material, finely ground glass (average particle size ~ 1 μm) and calcium oxide. Glass with dimensions of about 50–60 µm was used as a filler. As part of the work, it was shown that with an increase in the concentration of the binder (finely ground glass and calcium oxide) from 20 to 50%, the strength of the glass concrete composition increases from 35.7 to 110.3 MPa. Thermodynamic calculations and XRD analysis showed that hydrosilicate CaO*2SiO2*2Н2O is formed during hydration, which is the main carrier of the strength properties of mixtures. An important factor influencing the process of phase formation of CaO*2SiO2*2Н2O is the presence of NaOH in the glass. The results of DTA analysis and data of thermodynamic calculations show that an increase in temperature and pressure accelerates the process of phase formation of CaO*2SiO2*2Н2O, the formation of other phases of calcium hydrosilicates does not occur.

Reviewer 2 Report

Dear Editor: For the study to be accepted, the author (s) must respond to the following comments point by point to reach the level of a high-quality journal. 

1.     The followings are my comments regarding this manuscript. The authors should briefly discuss their innovation in the abstract, which has to be improved with more information.

2.     Missing the test procedure, explaining the calculations, and is disorganized. The abstract needs to be improved by highlighting this paper's key findings.The entire abstract section must be revised to briefly explain this research study's importance, investigations, and outcomes with advantages/significance.

Abstract :

3.     The entire abstract section must be revised to briefly explain this research study's importance, investigations, and outcomes with advantages/significance.  

4.     Resent a detailed graphical abstract for this work, which could be more interesting for the reader community. The novelty of the study should be reflected in the abstract.

Introduction: 

5.     The introduction section is not up to the mark. In the introduction section, you only need to connect state-of-the-art to your paper goals. Hence modify the entire section accordingly and present the specific goals/research objectives in the last part of the introduction section.

6.     The references in the text have to be listed in order from oldest to largest. Check them all.  

7.     MAJOR comment: The authors requested and must add more information and supported studies to the introduction since the introduction is poor and needs to be strengthened. Surprise !! The authors did not use the last decade's papers on the effect of temperature on the mechanical behavioure of cementitious materials, including concrete, and the mechanical behaviour of high-strength concrete!! The following papers published in reputed journals must be used in the introduction and the results in these papers have to be compared with the outcomes of this study to enhance the quality of the manuscript: 

Effect of temperature on the internal components including portlandite, weight loss, and compression stress-strain behavior of lime-based roof and screed paste; Effectiveness of Silicon Dioxide Nanoparticles (Nano SiO2) on the Internal Structures, Electrical Conductivity, and Elevated Temperature Behaviors of Geopolymer Concrete Composites; The influence of normal curing temperature on the compressive strength development and flexural tensile behaviour of UHPFRC with vipulanandan model quantification; Metamodel techniques to estimate the compressive strength of UHPFRC using various mix proportions and a high range of curing temperatures; Characterization and modeling the flow behavior and compression strength of the cement paste modified with silica nano-size at different temperature conditions

Methodology:

1.      Many grammatical errors need to be corrected. Several grammar errors can be observed in the paper, which is negatively affected by the paper's quality.

2.      The standard for all the tests must be mentioned in the study.

3.      Note that the laboratory conditions for these data sets differ based on various cases. There are two main challenges: first, the authors should validate and investigate different data and clarify how they have done it.

4.      The calibration steps of the compression machine with the properties of the machine have to be added to the manuscript.

5.               SEM photos have to be marked the Representative SEM images of asbestiform minerals of the materials. 

Results and discussion:

6.      The quality of figures are very poor, and the authors have to format the graphs to represent scientific data visually. They enable readers to visualize information that is often hard to grasp from the text. 

7.      Without witness lab photos, the results are not believable. The shape of the sample failures and the paths have to be provided in the study.

8.      The results are not repeatable? Error bars for at least three data sets for the same testing condition must be provided.

Conclusion (s):

1.     The conclusions are so poorly written. Please modify it to represent the outcomes of the study. This major deduction from this study does not demonstrate adequate uniqueness/novelty of the finding from this detailed research.

2.     Revision of the conclusions section is much required. It is not showcasing the entire essence of the detailed work presented in the paper. Also, inculcate the author's comments on the potential of the usage of graphene and its derivative with the comparison with potential alternatives in use for the current practice. This major deduction from this study, does not demonstrate adequate uniqueness/novelty of the finding from this detailed research.

Dear Editor: For the study to be accepted, the author (s) must respond to the following comments point by point to reach the level of a high-quality journal. 

1.     The followings are my comments regarding this manuscript. The authors should briefly discuss their innovation in the abstract, which has to be improved with more information.

2.     Missing the test procedure, explaining the calculations, and is disorganized. The abstract needs to be improved by highlighting this paper's key findings.The entire abstract section must be revised to briefly explain this research study's importance, investigations, and outcomes with advantages/significance.

Abstract :

3.     The entire abstract section must be revised to briefly explain this research study's importance, investigations, and outcomes with advantages/significance.  

4.     Resent a detailed graphical abstract for this work, which could be more interesting for the reader community. The novelty of the study should be reflected in the abstract.

Introduction: 

5.     The introduction section is not up to the mark. In the introduction section, you only need to connect state-of-the-art to your paper goals. Hence modify the entire section accordingly and present the specific goals/research objectives in the last part of the introduction section.

6.     The references in the text have to be listed in order from oldest to largest. Check them all.  

7.     MAJOR comment: The authors requested and must add more information and supported studies to the introduction since the introduction is poor and needs to be strengthened. Surprise !! The authors did not use the last decade's papers on the effect of temperature on the mechanical behavioure of cementitious materials, including concrete, and the mechanical behaviour of high-strength concrete!! The following papers published in reputed journals must be used in the introduction and the results in these papers have to be compared with the outcomes of this study to enhance the quality of the manuscript: 

Effect of temperature on the internal components including portlandite, weight loss, and compression stress-strain behavior of lime-based roof and screed pasteEffectiveness of Silicon Dioxide Nanoparticles (Nano SiO2) on the Internal Structures, Electrical Conductivity, and Elevated Temperature Behaviors of Geopolymer Concrete CompositesThe influence of normal curing temperature on the compressive strength development and flexural tensile behaviour of UHPFRC with vipulanandan model quantificationMetamodel techniques to estimate the compressive strength of UHPFRC using various mix proportions and a high range of curing temperaturesCharacterization and modeling the flow behavior and compression strength of the cement paste modified with silica nano-size at different temperature conditions

Methodology:

1.      Many grammatical errors need to be corrected. Several grammar errors can be observed in the paper, which is negatively affected by the paper's quality.

2.      The standard for all the tests must be mentioned in the study.

3.      Note that the laboratory conditions for these data sets differ based on various cases. There are two main challenges: first, the authors should validate and investigate different data and clarify how they have done it.

4.      The calibration steps of the compression machine with the properties of the machine have to be added to the manuscript.

5.               SEM photos have to be marked the Representative SEM images of asbestiform minerals of the materials. 

Results and discussion:

6.      The quality of figures are very poor, and the authors have to format the graphs to represent scientific data visually. They enable readers to visualize information that is often hard to grasp from the text. 

7.      Without witness lab photos, the results are not believable. The shape of the sample failures and the paths have to be provided in the study.

8.      The results are not repeatable? Error bars for at least three data sets for the same testing condition must be provided.

Conclusion (s):

1.     The conclusions are so poorly written. Please modify it to represent the outcomes of the study. This major deduction from this study does not demonstrate adequate uniqueness/novelty of the finding from this detailed research.

2.     Revision of the conclusions section is much required. It is not showcasing the entire essence of the detailed work presented in the paper. Also, inculcate the author's comments on the potential of the usage of graphene and its derivative with the comparison with potential alternatives in use for the current practice. This major deduction from this study, does not demonstrate adequate uniqueness/novelty of the finding from this detailed research.

Author Response

The introduction section is not up to the mark. In the introduction section, you only need to connect state-of-the-art to your paper goals. Hence modify the entire section accordingly and present the specific goals/research objectives in the last part of the introduction section.

Before:

Modern society produces a large amount of non-decomposing and hardly decomposing waste as waste, most of which accumulates and leads to an increase in the anthropogenic load on the environment. The most typical example is industrial and domestic glass waste [1,2]

After:

In the process of its development, mankind produces a large amount of non-decomposable and difficult to decompose materials. [Resource Recovery from Wastes: Towards a Circular Economy (ISSN) /  L. E Macaskie, D. J Sapsford, W. M Mayes// Royal Society of Chemistry; 1st edition (October 30, 2019) ISBN-13 ‏ : ‎ 978-1788013819] after the use of which a large amount of waste is generated that has a significant anthropogenic load on the environment [Zhen Zhang, Muhammad Zeeshan Malik, Adnan Khan, Nisar Ali, Sumeet Malik, Muhammad Bilal,  Environmental impacts of hazardous waste, and management strategies to reconcile circular economy and eco-sustainability, Science of The Total Environment, Volume 807, Part 2, 2022, 150856, ISSN 0048-9697, https://doi.org/10.1016/j.scitotenv.2021.150856.]. The disposal of these wastes in landfills is not rational from an economic point of view [Bo Wang, Chunyu Ren, Xiaoyang Dong, Bin Zhang, Zhaohua Wang, Determinants shaping willingness towards on-line recycling behaviour: An empirical study of household e-waste recycling in China, Resources, Conservation and Recycling, Volume 143, 2019, Pages 218-225, ISSN 0921-3449, https://doi.org/10.1016/j.resconrec.2019.01.005.], as it removes the material from economic circulation. The use of waste in the production of materials will reduce the anthropogenic load on the environment and reduce the cost of finished products. The most typical example is industrial and domestic glass waste. [Lanh Si Ho, Trong-Phuoc Huynh, Recycled waste medical glass as a fine aggregate replacement in low environmental impact concrete: Effects on long-term strength and durability performance, Journal of Cleaner Production, Volume 368, 2022, 133144, ISSN 0959-6526, https://doi.org/10.1016/j.jclepro.2022.133144., Li Pang Wang, Pin Wei Tseng, Kai Jyun Huang, Yan Jhang Chen, Foam glass production from waste bottle glass using silicon cutting waste of loose abrasive slurry sawing as foaming agent, Construction and Building Materials, Volume 383, 2023, 131344, ISSN 0950-0618, https://doi.org/10.1016/j.conbuildmat.2023.131344.]

Before:

In the Russian Federation alone, about 35–40 million tons of municipal solid waste are generated annually, while only 3–4% of MSW is recycled [3]. The amount of cullet for different territories of the country is 6 - 17 wt. % of the total mass of waste. The annual volume of cullet that ends up in solid waste landfills is 2–6 million tons. Compared to the annual need for building materials, this value is small, but it is necessary to take into account the environmental effect not only from the disposal of the solid waste component, but also the possibility of reducing the extraction of natural resources when replaced by raw materials of anthropogenic origin. In addition, the use of waste is 2-3 times cheaper than natural raw materials[4], fuel consumption when using certain types of waste is reduced by 10-40%, and specific investment by 30-50%. At the same time, the use of glass as a filler of concrete compositions is difficult due to the softening of the system during the chemical interaction of glass and cement, respectively, the strength of such a product is not high [Weiguo Shen, Mingkai Zhou, Qinglin Zhao, Study on lime–fly ash–phosphogypsum binder // Construction and Building Materials, Volume 21, Issue 7, 2007, Pages 1480-1485, ISSN 0950-0618, https://doi.org/10.1016/j.conbuildmat.2006.07.010., лодшл].

A large amount of industrial and domestic glass waste is generated annually in the world, which can potentially be used in the production of building materials and products. [Edward Harrison, Aydin Berenjian, Mostafa Seifan, Recycling of waste glass as aggregate in cement-based materials, Environmental Science and Ecotechnology, Volume 4, 2020, 100064, ISSN 2666-4984, https://doi.org/10.1016/j.ese.2020.100064.]. In addition, the use of waste is 2-3 times cheaper than natural raw materials[ Fakhratov M.A. Effektivnaya tekhnologiya ispolzovaniya promyshlennykh otkhodov v proizvodstve betona i zhelezobetona // Stroit. Materialy. 2003. № 12. С. 48-49], fuel consumption when using certain types of waste is reduced by 10-40%, and specific capital investments by 30-50%. At the same time, the use of glass as a filler for concrete compositions is difficult due to the softening of the system during the chemical interaction of glass and cement, respectively, the strength of such a product is not high [Weiguo Shen, Mingkai Zhou, Qinglin Zhao, Study on lime–fly ash–phosphogypsum binder // Construction and Building Materials, Volume 21, Issue 7, 2007, Pages 1480-1485, ISSN 0950-0618, https://doi.org/10.1016/j.conbuildmat.2006.07.010]. In general, the use of glass as a filler for various concrete compositions is a promising direction. [Xi Jiang, Rui Xiao, Yun Bai, Baoshan Huang, Yuetan Ma, Influence of waste glass powder as a supplementary cementitious material (SCM) on physical and mechanical properties of cement paste under high temperatures, Journal of Cleaner Production, Volume 340, 2022, 130778, ISSN 0959-6526, https://doi.org/10.1016/j.jclepro.2022.130778. найти и добавить еще статьи.].

Reducing the particle size leads to an increase in chemical activity. Accordingly, a high fineness of grinding increases the chemical activity of the system, which makes it possible to obtain high-strength material without the use of an autoclave.

The study is aimed at obtaining building materials from waste industrial and household glass.

As the main result of this work, it can be noted that the grinding of particles to a size of ~1 μm makes it possible to obtain a high-strength building material based on glass waste without the use of an autoclave.

This strength for the CaO-SiO2 system, without the use of an autoclave, has not been previously obtained.

Resent a detailed graphical abstract for this work, which could be more interesting for the reader community. The novelty of the study should be reflected in the abstract.

Переделал необходимые рисунки.

Remade the necessary drawings.

The references in the text have to be listed in order from oldest to largest. Check them all. 

Ссылки поменял и добавил, несколько более современных источников. Благодарю, за информацию. Добавил ссылки на перечисленные работы.

Links changed and added some more modern sources. Thank you for the information. Added links to the listed works.

The standard for all the tests must be mentioned in the study.

Для определения прочности и плотности использовались следующие стандарты:  C109 / C109M 13 Standard Test Method for Compressive Strength of Hydraulic Cement Mortars.

The following standards were used to determine strength and density: C109 / C109M 13 Standard Test Method for Compressive Strength of Hydraulic Cement Mortars.

No other mechanical studies have been carried out.

Note that the laboratory conditions for these data sets differ based on various cases. There are two main challenges: first, the authors should validate and investigate different data and clarify how they have done it.

Studies were carried out for 3 cases. 1) this is the hydration temperature of 40 C. Under normal conditions, hydration does not occur. 2) a temperature of 100 C simulates the steaming process 3) a temperature of 180 C and a pressure of 8 atm simulates the process of autoclave hardening. So I added suggestions:

At 20°C, the mixture did not set, so the temperature was raised to 40°C. The temperature of 100°C simulates the process of steaming in industrial steaming chambers.

These conditions simulate the process in industrial autoclaves.

The calibration steps of the compression machine with the properties of the machine have to be added to the manuscript.

Added information about the machine: Instron 3360 Series Testing Press (5 tons). Manufacturer information: Instron, USA. Purpose: Testing of building materials. Scope: Determination of strength characteristics of materials. Characteristics: Maximum load 5 tons.

If necessary, I can attach a certificate for verification of equipment. It's actually written in Russian.

SEM photos have to be marked the Representative SEM images of asbestiform minerals of the materials.

Need to think about what is representative.

The quality of figures are very poor, and the authors have to format the graphs to represent scientific data visually. They enable readers to visualize information that is often hard to grasp from the text.

Inserted other pictures for SAM.

Without witness lab photos, the results are not believable. The shape of the sample failures and the paths have to be provided in the study.

The results are not repeatable? Error bars for at least three data sets for the same testing condition must be provided.

Added a drawing with samples. (Fig. 4).

The sampling was carried out on 5 samples. Standard deviation error added.

Revision of the conclusions section is much required. It is not showcasing the entire essence of the detailed work presented in the paper. Also, inculcate the author's comments on the potential of the usage of graphene and its derivative with the comparison with potential alternatives in use for the current practice. This major deduction from this study, does not demonstrate adequate uniqueness/novelty of the finding from this detailed research.

We did not use graphene in our work. I wrote about graphene. Added a text to the reference: The use of nanoscale components can significantly improve the physical and mechanical characteristics of concrete compositions [Wengui Li, Fulin Qu, Wenkui Dong, Geetika Mishra, Surendra P. Shah, A comprehensive review on self-sensing graphene/cementitious composites: A pathway toward next-generation smart concrete, Construction and Building Materials, Volume 331, 2022, 127284, ISSN 0950-0618, https://doi.org/10.1016/j.conbuildmat.2022.127284.].

Round 2

Reviewer 1 Report

Dear Authors, 

Thank you for incorporating the changes and answering my questions. 

Reviewer 2 Report

Dear Editor: 

The authors carefully studied the reviewer's comments and revised the manuscript. In my opinion, this manuscript's quality meets the journal's requirements. I suggest this manuscript be accepted and published in this journal.

Dear Editor: 

The authors carefully studied the reviewer's comments and revised the manuscript. In my opinion, this manuscript's quality meets the journal's requirements. I suggest this manuscript be accepted and published in this journal.